# Machine Learning for Detection and Prediction of Crop Diseases and Pests: A Comprehensive Survey

Tiago Domingues [1,2,*], Tomás Brandão [1] and João C. Ferreira [1,2]

1    Instituto Universitário de Lisboa (ISCTE-IUL), ISTAR-IUL, 1649-026 Lisboa, Portugal
2    Inov Inesc Inovação, Instituto de Novas Tecnologias, 1000-029 Lisbon, Portugal
*    Correspondence: tards@iscte-iul.pt

**Abstract:** Considering the population growth rate of recent years, a doubling of the current worldwide crop productivity is expected to be needed by 2050. Pests and diseases are a major obstacle to achieving this productivity outcome. Therefore, it is very important to develop efficient methods for the automatic detection, identification, and prediction of pests and diseases in agricultural crops. To perform such automation, Machine Learning (ML) techniques can be used to derive knowledge and relationships from the data that is being worked on. This paper presents a literature review on ML techniques used in the agricultural sector, focusing on the tasks of classification, detection, and prediction of diseases and pests, with an emphasis on tomato crops. This survey aims to contribute to the development of smart farming and precision agriculture by promoting the development of techniques that will allow farmers to decrease the use of pesticides and chemicals while preserving and improving their crop quality and production.

**Keywords:** plant diseases and pests; classification; detection; forecasting; precision farming; machine learning; smart farming





## 1. Introduction

Due to extremely high infant mortality, the human population of the planet increased slowly until the year 1700. The first billion was reached in ca. 1800, followed by the second billion in 1928, the third billion in 1960. In 2017, the world's population reached its seventh billion. The fast population growth over recent decades is mainly due to better medical care. According to predictions from the United Nations, the world's population is expected to reach 9.7 billion in 2050, and 10.9 billion in 2100 [1].

Rapid population growth over recent decades has resulted in an increased demand for agricultural goods, which in turn has lead to a large expansion of cultivation [2]. To meet rising population demands for food, bio-fuels, and animal products, crop yield production must double its output by 2050. In order to achieve this goal, key crop yields must improve by 2.4% each year, but they are now only increasing by roughly 1.3% per year [3]. However, fulfilling this condition will have negative consequences for the ecosystem, including the loss of biodiversity and increased greenhouse gas emissions. Traditional agricultural production is not sustainable from an economic or environmental standpoint; hence, it is critical to optimize the use of resources such as water and soil to enable high yield crops [2].

Moreover, crop output is continually threatened by diseases and insect pests. It is estimated that between 20% to 40% of yearly crop production is lost due to plant diseases and insect assaults across the world, costing the global economy $220 billion and $70 billion, respectively. The amount of these losses varies across the globe and often occurs due to transboundary plant pests and diseases. For instance, the spread of crop pests and pathogens between 1950 and 2000 was greater in North America when compared with other world regions [4].

Pest damage and development are affected by the rise in global temperature brought by climate change. When the temperature rises, the metabolic rate of insects increases,

driving them to consume more food and inflict more damage. Growth rates of several insect species are also affected by temperature. For each degree of average global warming of the earth's surface, worldwide agricultural losses due to insect pests are expected to increase by 10% to 25% [5].

Pesticides and chemical treatments have long been used by farmers to keep pests away. The use of pesticides for crop protection is on the rise [6], with negative consequences for human health and increased environmental damage to soil and groundwater. On the other hand, this also increases the risk of pests developing pesticide resistances [5].

The traditional method of detecting and identifying plant diseases involves naked eye observation by experts. This takes time and talent, and is not a practical solution for monitoring large farms. Therefore, to overcome the limitations of manual detection, automated methods for crop monitoring and forecasting are required [7]. A system capable of performing such tasks can play an important role in avoiding the excessive use of pesticides and chemicals, reducing both the damage caused to the environment and the production costs associated with the use of pesticides and chemicals [7].

The growing availability of big data analysis methods has the potential to spur even more research and development in smart farming. Besides promoting higher yield crops in a more sustainable manner, it also aims to contribute to event forecasting, detection of diseases, and management of water and soil. Big data is coming to the agriculture domain by collecting data from meteorological stations, remote sensors, historical data, and publicly available data-sets [8].

ML approaches have been successfully utilized in a variety of areas, including illness detection from medical images [9], image classification on large data-sets [10], self-driving automobiles [11], and academic research fields such as physics [12].

ML-based applications for agriculture are still young, but are already showing promise. For instance, disease classification from images can be done using popular Convolutional Neural Network (CNN) architectures for different plants with different diseases [13]; relationships between weather data and pest occurrence can be retrieved using Long Short Term Memory (LSTM) networks for forecasting future pest attacks [14]; insect detection on leaves can be performed using object segmentation and deep learning techniques [15].

Commercial tools and services for smart farming that make extensive use of machine learning are currently available to farmers. A few examples are as follows. *Plantix*, created by the German startup *Progressive Environmental and Agricultural Technologies* (PETA), is an android-based farming assistant tool that provides crop health information, helping with identification of plant diseases using computer vision and deep learning techniques [16]. Other examples of similar applications are *Agrio* [17] and *CropDiagnosis* [18]. *Gamaya* is a startup company based on Switzerland that offers a wide variety of smart farming services services based on the analysis of images images acquired by drones connected to IoT systems [19]. The asian *iFarmer* [20] is another company that offers IoT-based soil analysis and satellite imaging-based crop monitoring solutions. *See & Spray*, developed by California-based *Blue River Technology*, is a large tow-behind herbicide sprayer, that uses computer vision and deep learning-based algorithms to automatically locate and identify weeds (in real time), applying herbicides to the specific locations found rather than to the entire field [21].

Some related surveys can be found in the literature, but most of them are focused on traditional ML techniques: in [22], a comparison of ML algorithms for predicting the yield of soybean crops is presented; in [23] research papers from the last ten years for predicting the start of disease at an early or presymptomatic stage are analysed and categorised; in [24], the possibility of using different ML techniques in agriculture are discussed, but most of the present work is about statistical forecasting methods from weather data for predicting wheat yield. Since the mentioned surveys do not simultaneously cover forecasting, detection, and classification of diseases and pests, and do not fully explore recent deep learning-based techniques, the review performed in this paper aims to fill the gaps on these subjects.

The literature review presented in this paper also aims to provide guidance on the development of such ML-based tools, in order to provide farmers with data-driven decision making assistance systems. In this way, farmers can be assisted with lowering the need for pesticide application and the harm that comes with it, while also preserving and enhancing crop quality and yield. This contributes to the continued availability of food to meet global population demands while doing less damage to the planet.

The application of ML-based techniques has promoted the emergence of projects that have enriched the development and the evolution of smart farming [25]. With this in mind, this article also contributes to the progression, development, and success of such projects.

## 2. Literature Review

Data gathering, data pre-processing (i.e., data preparation that includes feature extraction), and ML classification models are the three basic steps of ML applications, represented in Figure 1. The following sections present and discuss different approaches used in these three stages.



**Figure 1.** Simplification of the ML pipeline.

### 2.1. Data Acquisition

Data acquisition is the process of gathering data from various sources systems [26]. Previous studies gather their data various sources to be used for ML techniques. Some of them produce their own images by taking pictures of plants in greenhouses, such as in the studies from Gutierrez et al. [15] and Raza et al. [27]. However, image data acquisition using manual processes, as done by many, generally results in small image data-sets, which can compromise the development of effective ML-based models. Weather data collection is also proposed in the literature using for instance sensors in greenhouses, as done by Rustia and Lin [28]. Meteorological data can also be obtained from weather stations of regional areas, which typically store records for a longer period of time [14,29].

Images can be collected using search engines on their own [30,31]. This approach can get a large number of images, but ground truth must be checked by domain experts, and data cleaning is frequently used to filter out images that do not meet the requirements.

Remote sensing images from satellites and drones have the advantage of being able to retrieve image data for large agricultural areas. Remote sensing data from satellites typically consists of multi-temporal and hyper-spectral imagery data, which can be used to assess the development of the crops. This task can be performed by monitoring the evolution of vegetation indices [32], which provide important information about the development status of the crop fields. Spectral imagery can be used for computing different vegetation indexes, such as those proposed in [33–39], which are robust to variations on the sun illumination [37], an important advantage when compared to visible light spectrum imagery.

Images retrieved from drones can also be used, but have additional needs: to define the path of the device; to coordinate the drone position with the camera for image acquisition; and to correct geometric distortions on each acquired image in order to merge the different acquired images in order to reconstruct a larger image of the whole field [40].

Therefore, it can be stated that data consists of different modalities and variables. With ML-based and data analysis techniques it can be possible to understand their interaction and how they relate to a studied outcome. In the context of the cultivation fields, the questions are usually: which disease is affecting crops? What pest is causing damage? What is the relation between weather data and disease and pest occurrence? The most

important variables related to the appearance of plant diseases and pests are reviewed in Section 2.1.1.

Freely available data-sets can also be utilised for the development of ML-based applications. This enables researchers to directly compare the performance of different ML techniques and approaches. In Section 2.1.2, a brief summary of the main available data-sets is presented.

The data conditions a significant impact on the performance of ML models. Section 2.1.3 addresses this issue. Data-sets should be representative and include enough records for the model to perform an effective generalization.

### 2.1.1. Variables Influencing Crop Diseases and Pests

It's crucial to be able to predict the arrival of diseases and pests in crops, in addition to correctly detecting and identifying them. Real-time meteorological data obtained by unmanned observation planes, as well as long-term data analysis from weather stations, have been used to create models capable of anticipating disease occurrence. In [41], the General Infection Model, proposed in [42], was used for assessing the prediction capabilities of the system. It was found that, if integrated systems such as this are implemented and various input data-sets essential for interrelationship analyses are collected, accurate plant disease prediction systems can be constructed.

When it comes to forecasting occurrences, it's crucial to know which variables will have an influence on what is being forecast. In the work by Henderson et al. [43] this was done by discovering which weather variables influence the forecast. On the other hand, Lasso et al. [44] determined the time period window for each weather variable and crop-related feature that is the most significant for the appearance of coffee leaf rust disease in coffee crops.

In [45], Small et al. used weather data, information on potato and tomato crops resistance to late blight (from published literature and field experiments), and management strategies, to create a web-based decision support system that allows the dynamic prediction of disease outbreaks, with an emphasis on the late blight disease on tomato and potato crops.

The work proposed by Ghaffari et al. [46] addresses the very early detection of diseases in tomato crops using atmospheric data and volatile organic compounds. Plants produce a wide spectrum of volatile organic compounds in reaction to physical and biotic stress, as well as infection [47]. In [46], the diseases under study were the powdery mildew and spider mites.

A model developed by Diepeveen et al. in [48] can be used in agriculture to understand the influence of location and temperature on crops. In addition, elements such as soil, humidity, rainfall, and moisture were found to have an influence on crop yield [49].

Plant diseases and pest development are greatly influenced by weather and environment conditions [50]. Humidity is a favorable condition for the development of fungus diseases. The humidity can be caused by the weather or by poor watering practices that cause a high wetness among the leaves, making tomatoes more susceptible to diseases, e.g., leaf mold or bacterial spot [51].

In addition, temperature is a primary driver of insect development, affecting their metabolic rate and population growth [5].

Plants absorb part of the radiation coming from the sun and reflect the rest. Depending on the health of the plant, the amount of radiation absorbed and reflected differs. This difference can be used to distinguish between healthy and diseased plants and to assess the severity of the damage [52]. The concept is illustrated in Figure 2.

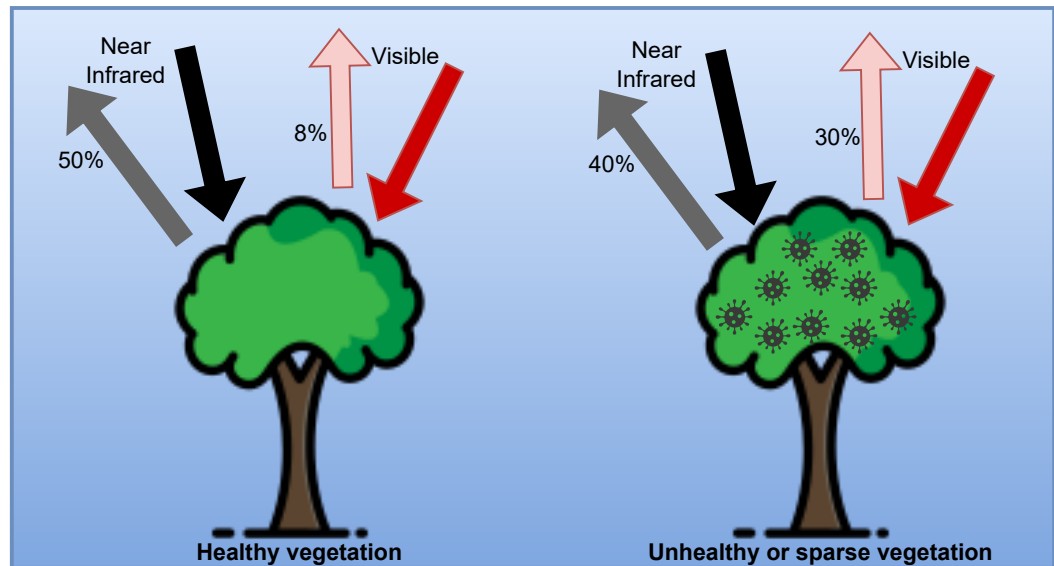

**Figure 2.** Absorbed and reflected radiation for plant's health estimation (adapted from [53]).

Temperature

Insects are ectothermic, meaning that they cannot regulate their internal temperature and have to rely on environmental heat sources. Temperature affects the population growth and metabolic rates of insects [5]. Thus, the duration of an insect's life cycle is highly influenced by the number of days where the temperature is suitable for its development. Two temperature thresholds can be define: an upper threshold, in which insect development slows down or stops and a lower one where there is no insect growth. These thresholds vary according to the specific insect species.

*Degree day* is a concept concerning the accumulation of heat by insects [54]. One degree day is a period of 24 h in which the temperature was one degree above a given baseline. Different models for determining the number of degree days associated to common pest species were proposed in [55]. For instance, tomato crops are susceptible to the greenhouse white fly (*Trialeurodes vaporariorum*), whose number of degree days from egg to adult is 380 DGG [56]. Depending on the temperature of the environment, this development time can be longer or shorter.

*Biofix date* is the date to start accumulating degree days associated with a given insect species [57]. This date can be determined by noticing specific insect species on traps or by detecting eggs on plant leaves. From this date, degree days can be used to estimate the period at which insects are reaching a given development stage suitable for pesticide application. Temperature and weather forecasts are nowadays sufficiently accurate to enable the estimation for the time required for an insect to reach a given development status [58].

In the context of ML-based applications, related work focused on studying the impact of weather in pest insect development found a higher correlation between the number of pest catches and temperature, when compared with other factors [28,32].

Some diseases affect the transpiration rate of the plant and, consequently, its temperature [27]. Therefore, plant leaf temperature can be used for disease detection. ML models can achieve higher accuracy for disease identification when combining thermal images with visible light images. The benefits are more useful for early detection when the plant has not yet developed symptoms recognizable by the naked eye.

Humidity

Diseases affecting plants are often caused by fungus or bacterial pathogens. High relative humidity environments favor the development of these microorganisms. Thus,

humidity has to be managed by good watering practices, while avoiding excessive leaf moisture and soil moisture [59].

Different studies using regression models and weather data demonstrate the influence of humidity on disease and pest development [14,29]. Thus, the collection of humidity records in greenhouses using sensors can be helpful for disease forecasting.

Leaf Reflectance

Plants absorb solar radiation between 400 to 700 nm (photosynthetically active radiation) which corresponds approximately to the visible light region. For wavelengths greater than 700 nm (red) in the Near Infra-Red (NIR) region there is a sharp order-of-magnitude increase in leaf reflectance due to chlorophyll characteristics, a phenomenon known as *red edge* [60].

Diseased plants with damaged leaves have different leaf spectral reflectance compared to a healthy plant because of the different chlorophyll concentration and leaf tissue damage. Diseased plants end up absorbing less of the visible light and more of the NIR light. From this knowledge, disease detection can be done using leaf reflectance information [40,52,53]. In a study concerning late blight infection, a disease that tomatoes are also susceptible to, it was found that spectral differences in the visible region between healthy and diseased plants are small and more significant differences are noticeable in the NIR [40].

Various vegetation indices can be retrieved from remote sensing [33]. A common index is the Normalized Difference Vegetation Index (NDVI) (Figure 2) for assessing the degree of vegetation of an area by using leaf reflectance information. NDVI can be computed using satellite data or from modified cameras [40,53]. It was found that the combination of NDVI and temperature gives higher accuracy in predicting pests appearances than weather variables alone [32]. NDVI can also be used as input data for ML models to accurately evaluate disease severity.

Pest development varies depending on the development stage of the plants. NDVI can be used to monitor plant growth and establish relationships between the crop stage development and pest occurrence.

2.1.2. Agriculture Data-Sets

Many data-sets used in the context of agriculture include images of plant diseases or pests with the goal of classifying them. *PlantVillage*, *PlantDoc*, *IP102*, *Flavia* and, *MalayaKew Leaf* are some data-sets that are freely available. Here is a brief summary of each of these:

- *PlantVillage* [61]: popular data-set used for plant disease classification. Specifically for tomato, it contains 18,160 images representing leaves affected by bacterial spot, early blight, late blight, leaf mold, septoria leaf spot, spider mites, two-spotted spider mite, target spot and tomato yellow leaf curl virus. It also includes images of healthy leaves. Figure 3 depicts two sample images taken from this data-set.
- *IP102* [62]: data-set for pest classification with more than 75,000 images belonging to 102 categories. Part of the image set (19,000 images) also includes bounding box annotations. This is a very difficult data-set because of the variety of insects, their corresponding development stages (egg, larva, pupa, and adult) and image backgrounds. The data-set is also very imbalanced. Figure 4 presents two examples of images from this data-set.
- *PlantDoc* [63]: contains pictures representing tomato diseases which were acquired in the fields. Among the considered diseases are: tomato bacterial spot, tomato early blight, tomato late blight, tomato mold, tomato mosaic virus, tomato septoria leaf spot, tomato yellow virus and healthy tomatoes.
- *Flavia* [64]: contains photos of isolated plant leaves over a white background and in the absence of stems. This data-set covers 33 plant species.
- *MalayaKew Leaf* [65]: was gathered in England's Royal Botanic Gardens at Kew. It contains images of leaves from 44 different species. There are situations where leaves from

different species are very similar, presenting a greater challenge for the development of plant identification models.

Tomato Powdery Mildew Disease (TPMD) is a different type of data-set because it is related to meteorological data. It offers statistics on powdery mildew disease susceptibility depending on a variety of weather-related variables such as humidity, wind speed, temperature, global radiation, and leaf wetness [66].

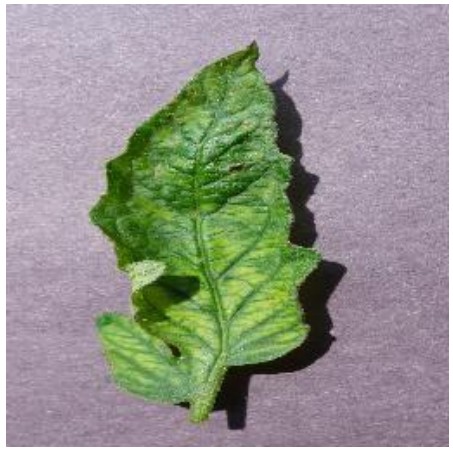

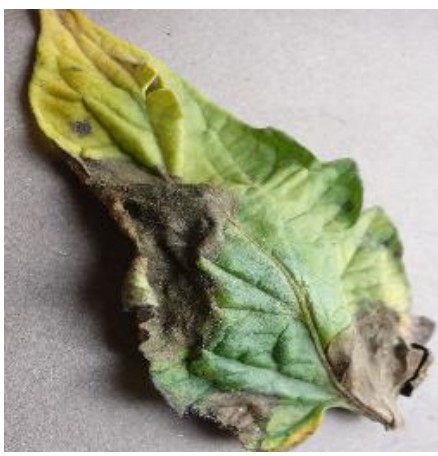

(**a**) Tomato leaf affected by mosaic virus disease　　　(**b**) Tomato leaf affected by late blight disease

**Figure 3.** Examples of tomato leaves affected by diseases taken from the *PlantVillage* data-set [61].

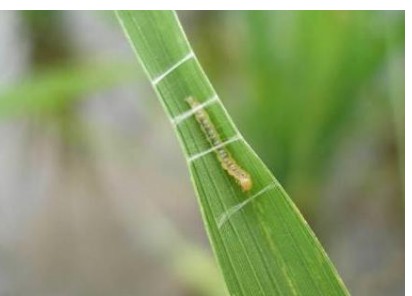

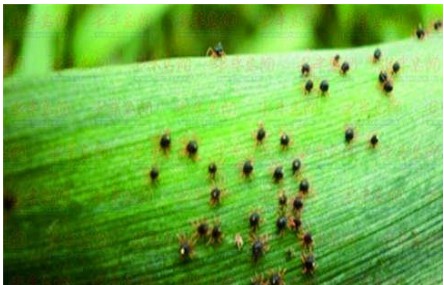

(**a**) Rice leaf roller (*Marasmia exigua*)　　　　　　(**b**) Winter grain mite (*Penthaleus major*)

**Figure 4.** Examples of insect images taken from the *IP102* data-set [62].

### 2.1.3. Field-Collected vs. Laboratory-Collected Data

ML models performance is influenced by the quality and type of input (image or other). Images acquired in a controlled laboratory environment and images acquired in the field can result in completely different processes and/or results. The difficulty for disease and pest classification is much higher for images acquired in the field than for images taken in a controlled environment.

Under a controlled laboratory environment, images typically contain a single leaf over a neutral artificial background [67]. The *PantVillage* data-set is an example of such situation [61]. It is possible to achieve great performance on these data-sets [13]. However, the creation of these types of data-sets is a time consuming and costly process.

When compared with images acquired in the laboratory, field images have much higher complexity, due to the presence of multiple leaves in the same image, presence of other plant parts, different shading, and lighting conditions, different ground textures, different backgrounds, etc. [63]. According to the studies in [63,68], training ML models using laboratory images provides poor outcomes when tested in the field, making them useless for the task. Training on field photographs and testing on laboratory photographs, on the other hand, produce reasonable outcomes [68]. The addition of field images in the training data has been shown to boost the results significantly, however testing on images

from alternative data sources is advised [68]. *PlantDoc* demonstrates that cropping the leaves improves the accuracy of CNN architectures when dealing with in-field photos [63].

Table 1 shows the performance achieved on a few studies that analysed the impact of image acquisition conditions on the performance of disease classification models. In each table cell, "L" corresponds to lab images, "F" to field images and "L + F" for both types of images. In addition, the data-sets associated to the weights of the pre-trained models that were used for Transfer Learning (further explored in Section 2.3.4) are also shown.

**Table 1.** Performance comparison of field vs. laboratory data.

| Study | Pretrained Weights | Training | Testing | Performance |
|---|---|---|---|---|
| [68] | - | L | F | 33.0% acc. |
| | | F | L | 65.0% acc. |
| | | L + F | L + F | 99.0% acc. |
| [63] | ImageNet | L | F | 15.0% acc. |
| | ImageNet + PlantVillage | F | F | 30.0% acc. |
| | ImageNet + PlantVillage | F (cropped images) | F (cropped images) | 70.0% acc. |
| [13] | ImageNet | L | L | 99.0%+ acc. |

### 2.2. Data Pre-Processing

Pre-processing data before feeding it to the model is common in most ML-based applications. Images are typically pre-processed using computer vision techniques to remove noise, to enhance the image contrast, to extract the regions of interest, to extract image features, etc. In general, image pre-processing steps usually lead to better model outcomes. The most common data pre-processing techniques are covered in the following sub-sections.

#### 2.2.1. Noise Reduction

Different types of filters, such as Gaussian and median filters, are used to reduce noise to obtain smoother images. These filters have an effect of blurring and removing non relevant details of an image, at the expense of potentially losing relevant textures or edges [69].

Erosion and dilatation are two morphological image operations that can be applied to binary or grey-scaled images. Erosion removes islands and tiny items, leaving only larger objects. In other words, it shrinks the foreground objects. On the other hand, dilation increases the visibility of items and fills in tiny gaps, adding pixels to the boundaries of objects in an image [70]. These operations reduce details and enhance regions of interest. These methods are helpful, for instance, for pest detection against a neutral background, such as images of traps with captured insects [28,71].

Images are usually stored in the RGB format, which is an additive color model of red, green, and blue components. Due to the high correlation between these color components, it is usually not suitable to perform color segmentation in the RGB color space. Therefore it is important to bear in mind that there are others color spaces such as HSV or L*a*b*. In HSV the color components are: hue (pure color), saturation (shade or amount of grey), and value (brightness). In the L*a*b* color space, L* is the luminance (brightness), a* is the value along the red-green axis, and b* is the value along the blue-yellow axis. In these color spaces, the brightness of a color is decoupled from its chromaticity, allowing the images to be processed with different lighting conditions [69]. This is significant in the context of agricultural images acquired in the fields, since they can have been shot under various lighting circumstances or at different times of the day.

Histogram equalization is a technique for adjusting contrast. In low contrast images, the range of intensity values is smaller than in high contrast images. Equalization of the

histogram spreads out the intensity levels throughout values in a wider range. Contrast enhancement is not directly applied in the RGB color space, because it applies to brightness values. Thus, images have to be converted either to grey-scale or to a color space that contains a brightness component, such as the HSV or L*a*b* color spaces [69].

### 2.2.2. Image Segmentation

Image segmentation is the process of grouping pixels into regions of interest. In the context of crop disease identification, these regions of interest can be, for instance, diseased areas on the plant leaves, for assessing the severity of the infection by the amount of the infected area, or for background removal, since the removal of the background allows highlighting of the regions of interest for further analysis. An example of background removal is shown in Figure 5.

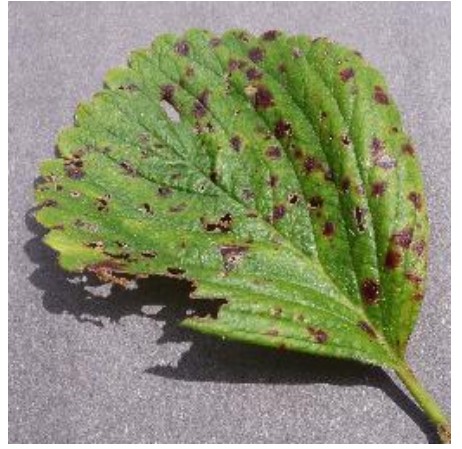
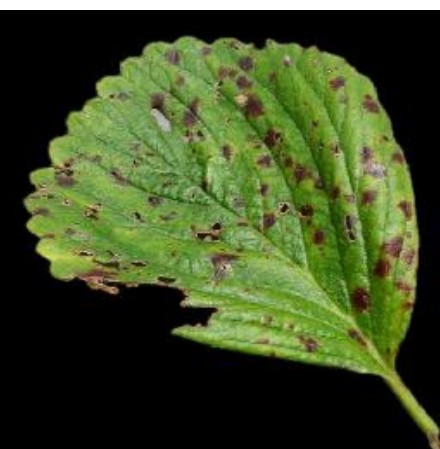

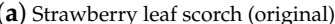

(**a**) Strawberry leaf scorch (original)          (**b**) Strawberry leaf scorch (segmented)

**Figure 5.** Example of background removal from the *PlantVillage* data-set [51].

Blob detection is a computer vision technique for getting regions of pixels that share common properties. The properties of these regions, such as color and brightness, differ greatly compared to their surroundings. This technique can be used, for instance, to detect and count insects in images [28,71].

The k-means clustering algorithm is a popular unsupervised ML algorithm that can be used for image segmentation. Pixels are grouped into clusters which have pixels with similar color and brightness values. This technique is helpful, for instance, to detect damaged regions on leaves [31,72]. Fuzzy c-means is a soft clustering technique where a pixel can be assigned to more than one group. This method was used by Sekulska-Nalewajko and Goclawski [73] and Zhou et al. [74] for plant disease classification.

Region growing is a region-based image segmentation technique used by Pang et al. in [75] to accurately define the image regions corresponding to the plant leaf parts affected by disease.

Intensity thresholding is a straightforward and simplified approach for image segmentation. According to the pixel value, that pixel is classified into a group (e.g., healthy or diseased). When using this technique, images are frequently converted to grey-scale first and then thresholded using a grey intensity value [76]. Figure 6 shows an example of an image converted to grey-scale.

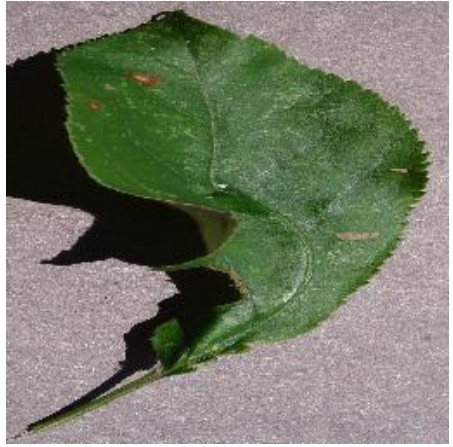 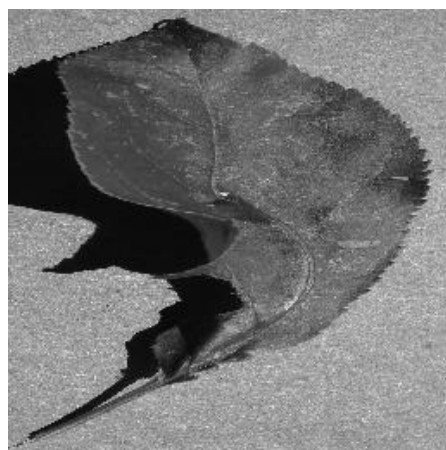

(**a**) Black rot apple (original)     (**b**) Black rot apple (grey)

**Figure 6.** Example of an image converted to grey-scale from the *PlantVillage* data-set [51].

### 2.2.3. Feature Extraction

Feature extraction is a common step in the pre-processing of images for shallow ML models. Common image feature extraction algorithms include the Histogram of oriented Gradient (HoG), Speeded Up Robust Features (SURF) and Scale Invariant Feature Transform (SIFT) [62,77]. Different feature extractors obtain different features that can be more or less suitable for the specific problem at hand. HoG focuses on the structure and shape of the image objects, by detecting edges on images oriented according to different directions. The distribution of gradients according to these directions are used as features. SIFT finds scale and rotation invariant local features through the whole image, obtaining a set of image locations referred to as the image's key-points. SURF is conceptually similar to SIFT, with the advantage of being much faster, which can be relevant for the implementation of real-time applications.

The distribution of image colors is represented by a color histogram. Since most diseases have symptoms that impact the color of the leaves, the histogram can also be used for distinguishing between healthy and unhealthy plants [77].

Some computer vision algorithms for feature extraction demand that pictures are converted to grey-scale, such as Haralick texture [78] or edge detection algorithms [79], etc. Haralick texture features are computed from a Grey Level Co-occurrence Matrix (GLCM), a matrix that counts the co-occurrence of neighboring grey-levels in the image. The GLCM acts as a counter for every combination of grey-level pairs in the image. Diseased and healthy leaves have different textures since a diseased leaf has a more irregular surface and a healthy leaf has a smoother one. These features allow differentiation of a healthy leaf from a diseased one.

Local Binary Pattern (LBP) [80] is another technique used for image texture features extraction robust to variations on lighting conditions. The LBP technique was used by Tan et al. in [81] for the extraction of information about diseases on tomato leaves.

Multi-spectral image data-sets can be exploited to create new data and improve the performance of models. For instance, in [40], originally, there were NIR pictures of the fields and from this data the authors created new images from spectral differences (between green and blue bands, and between NIR and green bands), band ratios and dimension reduction using principal component analysis. The authors also assess which type of data achieves best performance on the models.

### 2.2.4. Cropping and Resizing Images

Cropping and resizing images is used for decreasing the input image dimensions, to allow greater processing speed or to fit hardware requirements. It can also be used for

creating more data to train the models, for example, from a low number of high resolution pictures, a much higher number of low resolution images can be retrieved [40].

### 2.2.5. Pre-Processing in Tabular Data

Tabular data consisting of weather records was commonly found in the literature analysed in the scope of this paper. When gathering data records with varying dates and locations, these records can be integrated in two ways: cross-year, where models are validated over the years at the same location, and cross-location, where models are validated across the various locations for the same year. The average coefficient of determination ($r^2$) was found to be higher for cross-year models for all ML algorithms tested [29].

Common procedures in pre-processing are scaling/standardization of data and missing values processing [14]. Most algorithms require that there are no missing values in data and others, such as neural networks, can benefit from the normalization of feature values to improve training and reduce the effects of vanishing gradients [29].

Down sampling is a useful way to process data when there is a high number of records. In [52], measurements of leaf reflectance were done, from 760 to 2500 nm with a 1 nm interval. The 1740 wavelengths measurements were compressed into 174, and afterwards 10 wavelengths were selected using the stepwise method. From the regression analysis, results showed a coefficient of determination $r^2 = 0.94$ for these wavelengths and leaf severity. Experiments showed that fewer than those 10 wavelengths would worsen performance.

### 2.2.6. Pre-Processing in Deep Learning

Deep learning pre-processing does not focus on feature extraction since one of the most essential and beneficial properties of deep learning is its ability to generate features autonomously. For this reason, pre-processing is focused mainly on creating more images through data augmentation and resizing the input images to fit the models input parameters.

Some studies have compared the manual selection of features with deep learning. When it comes to categorizing insects in the field, manually selected features were not able to capture all of the relevant information about insect infestations or to handle the noise of real-world photos. Manually selected features were also not able to capture subtle differences between different insect species that share similar appearance [62]. For insect detection, deep learning techniques achieved higher accuracy and took less time to process since they efficiently select regions of interest [15]. In the work done by Brahimi et al. in [67], tomato disease classification using deep learning achieves higher accuracy, with values above 98%, but the accuracy of models using feature extractors is not very far behind, reaching values above 94%.

When comparing the use of original color pictures with images converted to grey-scale or background segmentation, deep learning models performed better in the original color pictures [13]. These findings are also confirmed in [82], where the performance of color vs. grey-scale pictures is compared. This supports the idea of deep learning not requiring extensive pre-processing of images. Nevertheless cropping images achieve better performance on field images classification, by increasing the region of interest and reducing the varying background [63].

Data augmentation is a process to artificially expand and increase the diversity of the training data-set. This process benefits the performance of the models, by introducing variability in the data and allowing a better generalization of the domain [83]. Some common transformations are rotation, cropping, scaling, and flipping.

Data cleaning is the process of assessing the quality of the data and to either modify or delete it. It is usually applied in studies that retrieve their data-set images from search engines in an automatic way, removing pictures that do not correspond to the intended labels or that do not comply with minimum resolution requirements [30,62].

Image resizing is usually performed to fit the input parameters of the models. Studies have compared the performance of the models with different input image sizes, and con-

cluded that with larger images the models achieve higher accuracy but require more time for each training epoch [68] and more powerful hardware [71].

Table 2 shows the pre-processing techniques applied to deep learning classification models analysed in the scope of this review. The 'type' column shows the data pre-processing technique used and the 'info' column contains additional details about it.

**Table 2.** Pre-processing when deep learning techniques were used.

| Study | Type | Info |
|---|---|---|
| [13] | Greyscale | - |
| | Background Segmentation | Masks |
| | Resize | 256 × 256 |
| [30] | Data Augmentation | Affine, perspective, rotation |
| | Data Cleaning | - |
| | Resize | 256 × 256 |
| [71] | Resize | 52 × 52, 112 × 112, 224 × 224 |
| [62] | Data Cleaning | - |
| | Resize | 224 × 224 |
| [15] | Data Augmentation | Crop, rotation, Gaussian noise, scale, flip |
| | Resize | 600 × 1024, 300 × 300 |
| [67] | Resize | 256 × 256 |
| [68] | Resize | 256 × 256 |
| [82] | Greyscale | - |
| | Resize | 60 × 60 |

From the table, it is noticeable that all analysed papers employing deep learning-based techniques used image resizing. It is also worth mentioning that the application of data augmentation was found in 25% of the depicted works, and the same goes for image color conversion to grey-scale and data cleaning.

*2.3. Machine Learning Models*

ML models enable researchers to get insight into data and existing correlations between various factors that influence occurrence of diseases and pests in crops. After data is processed and features are extracted, models can be used for classification, regression, among other goals. In classification, a new data sample is assigned a label according to the relations retrieved during the training process. In regression, a continuous output value is estimated from the input variables.

The following sub-sections contain a description about the ML models used, published work that have used them and the achieved performances. In addition, as a consequence of the conducted research, it was decided to include a sub-section about the use and potential of Transfer Learning (TF) in the research under consideration.

2.3.1. Support Vector Machine

SVM [84] is a model that creates a hyper-plane that separates two classes (can also be adapted and applied for multi-class problems). By maximizing the distance, or margin, between the nearest data points (support vectors) of each class to the hyper-plane, SVM chooses the optimum hyper-plane to segregate the data. SVM can also perform well in non-linear data by using the so called *kernel trick* technique. The SVM kernel is a function that transforms a low dimensional input space into a higher dimensional space that is linearly separable. For this reason, SVM can be very effective in high dimensional spaces.

SVM can also be used for regression problems [29,40,85]. Furthermore SVM can also be used in a hybrid way as Bhatia et al. did in [86], by using SVM together with logistic regression algorithm to predict powdery mildew disease in tomato plant.

A syntheses of agricultural studies using SVM as the ML model can be observed in Table 3. The type of SVM used, as well as its kernel and result can be observed. Linear, polynomial, and RBF kernels seem to be most commonly used on SVM-based classification and regression algorithms applied to agriculture contexts.

**Table 3.** SVM performance.

| Study | Classification/ Regression | Kernels | |
|---|---|---|---|
| | | Type | Results |
| [52] | Classification | Polynomial | 90.0% acc. |
| | | Radial Basis Function | 97.4% acc. |
| [29] | Regression | Not specified | SVM outperformed |
| [40] | Regression | Linear | $r^2 = 0.45$ |
| [27] | Classification | Linear | 90.0%+ acc. |
| [31] | Classification | Radial Basis Function | 90.5% acc. |
| | | Quadratic | 92.0% acc. |
| | | Linear | 91.0% acc. |
| | | Multi-Layer Perceptron | |
| | | Polynomial | |
| [67] | Classification | Not specified | 94.6% acc., 93.1% f1 |

SVM can achieve better performance than other ML techniques such as ANNs and conventional regression approaches in forecasting plant diseases [29].

### 2.3.2. Random Forest

Random Forest (RF) is a widely known ensemble built from decision trees trained on different subsets of the training data. Also, when deciding which variable to split on a node, RF considers a random set of variables and not the whole set of features. During classification, each tree votes and the class most agreed upon is returned. As each tree trained on a subset of data and of features, the computation is fast. A high number of trees and the diversity of each of them makes them robust to noise and outliers. Some studies that have employed Random Forest (RF) are shown in Table 4.

**Table 4.** Performance of Random Forests.

| Study | Classification/Regression | Number of Trees | Performance |
|---|---|---|---|
| [40] | Regression | 100 | $r^2 = 0.75$ |
| [32] | Regression | 200 | $r^2 = 0.75$ |
| [77] | Classification | - | 70.0% acc. |
| [67] | Classification | - | 95.5% acc., 94.2% f1 |

RFs can achieve greater accuracy with less number of samples when compared to other ML techniques [77].

### 2.3.3. Artificial Neural Networks

Artificial Neural Networks (ANN) are models inspired by biological brains. ANN consists of neurons distributed in input, hidden, and output layers and can have multiple hidden layers and multiple units in each layer. With more hidden layers, an ANN is able to

learn complex relations from the hierarchical combination of multiple features, and thus create high-order features, Figure 7 shows an illustration of an ANN. Deep learning is associated with ANNs that contain a large amount of layers.

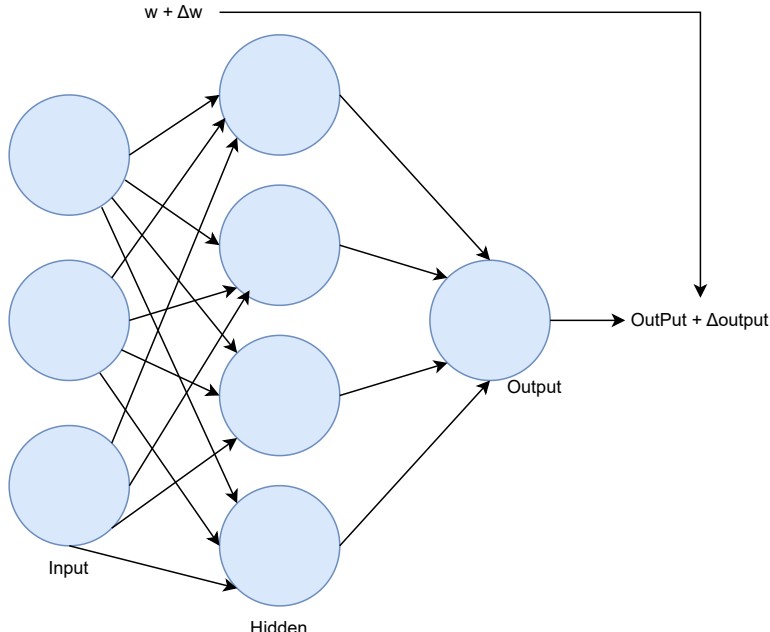

**Figure 7.** ANN example.

Learning occurs by a process called optimization, which is an iterative method for minimizing an error function, typically based the Gradient Descent algorithm. Instead of calculating the gradient from the entire data-set, the optimization process typically uses chunks of data records called batches. After the network processes the input, the output is compared to the expected output and the error is computed. The error is then propagated back through the network, one layer at a time, and the weights are updated according to the amount they contributed to the error. This updating process is called back-propagation. After all records in the data-set are processed once, a training epoch is completed. Training the network can require several epochs until desired results are achieved.

CNNs are a type of a deep learning network that commonly are applied on image classification tasks. In this type of network, the use of the so-called convolutional layers enables an hierarchical extraction of features, where simpler features such as edges are extracted in the first layers and more specific and complex features are extracted in deeper layers. The dimensionality of the input is decreased by the use of pooling layers. Fully connected neural networks are usually placed on top after the convolutional and pooling layers and act as classifiers using these high-level features.

Recurrent Neural Networks (RNN) are also a type of deep learning network, usually applied to time series data. RNNs extract features automatically from data and can capture temporal relationships. Because of the architecture of these networks, the gradients calculated to update the weights can become unstable, becoming too high (Exploding Gradient) or too low (Vanishing Gradient).

The recurrent layers can be structured in a wide variety of ways to produce distinct RNNs [87]. The LSTM cell was proposed by Hochreiter and Schmidhuber in [88]. Here, the remembering capacity for the standard recurrent cell was improved in order to deal with undesirable dependencies on the long-term.

Recently, Xiao et al. suggested in [14] that LSTM networks have specific advantages in processing time-dependent problems. LSTM networks can be used, for example, to retrieve relationships between meteorological data and pest occurrence in order to forecast future pest attacks.

In the context of agriculture, obtaining a large amount of annotated data for the training of ML-based algorithms can be a rather difficult task. Few-shot learning approaches have been trying to mitigate this problem by managing to learn with fewer data. The methods typically associated with this technique can be organized according to four groups [89]: data augmentation, metric learning, external memory, and parameter optimization. Yang et al. present a survey on the developments, application, and challenges of this approach.

When using ANN models, authors might use one of two methods. They either create their own model designs or adopt well-known architectures that have been shown to perform well in previous studies, particularly CNN architectures for image classification.

User-Defined Network Architectures

This sub-section presents studies where the authors defined their own neural network architectures.

In [40], Duarte-Carvajalino et al. built and compared the outcome of two different neural networks models. The first model was a Multi-Layer Perceptron (MLP) with 2 hidden layers, each having half the number of nodes of the previous layer. The authors used a learning rate of 0.01, the Adamax optimizer, batch normalization and dropout with probability of 0.2 in all layers, and ReLU as activation function. The other model was a CNN trained using the same hyperparameters used on the MLP. The CNN consists of two convolutional layers using 20 filter kernels of size $3 \times 3$, followed by a max pooling layer of size $2 \times 2$. The succeeding network layers are another two convolutional layers using 40 filter kernels of size $5 \times 5$ followed by a max pooling layer. After flattening, a dense layer is added before the output is computed. It was concluded that the CNN achieved better results than the MLP.

In [14], an LSTM network was used for processing time series data, i.e., winter and autumn data. The LSTM network consisted of two fully connected layers with five hidden units each. The results showed that the LSTM network achieved the best performance with 92% accuracy when compared to RF, SVM, and K-Nearest Neighbors (KNN). The Apriori algorithm [90] was applied for interpretability.

Disease prediction for different regions was also studied with the use of an ANN in [29]. In this case, the back-propagation neural network [91] and the generalized regression neural network [92] models were used.

A model suggested by Patil and Kumar in [93] attempted to identify the link between weather variables and the emergence of 4 types of rice diseases. In this work, the authors used an ANN to perform the detection, identification and prediction of the appearance of diseases in rice crops. The meteorological data-set referred to data between 1989 and 2019. The ANN consisted of 8 neurons in the input layer, 15 in the 2 hidden layers, and 5 in the output layer.

In [94], Sharma et al. performed a prediction of the potato late blight disease based on meteorological data only, using an ANN. In this case, data from 2011 and 2015 was used. Several tests with different network activation functions and data-set splits were done. It was concluded that the larger the data-set, the better was the performed prediction.

In addition, other algorithms relying on meteorological data and ANNs for performing predictions have been proposed. In [95], Dahikar and Rode present an ANN for predicting which crop will grow best in a certain area. The predictions were based on weather and soil data. Refs. [96,97] proposed ANN-based models for predicting crop yield.

Convolutional Neural Network Architectures

Image classification has achieved great results, with various model architectures being developed over the last 10 years. Most of these deep learning models were proposed in the context of the "Large Scale Visual Recognition Challenge" (ILSVRC). These models include well-known architectures such as AlexNet, GoogleNet, VGG, and ResNet, which have been widely used for image classification in different application domains.

Table 5 summarizes a set of studies that used pre-existing CNN architectures, depicting the architecture used in their work and the corresponding results.

**Table 5.** Performance of CNN architectures.

| Study | Architecture | Results |
|---|---|---|
| [13] | GoogleNet | 99.3% |
| | AlexNet | 99.3% |
| [30] | CaffeNet | 96.3% |
| [71] | VGG16 | 98.0% validation, 81.0% in new apple orchard |
| [62] | GoogleNet | 43.5% acc., 32.7% f1 |
| | FPN | 54.9% mAP 0.5 |
| | ResNet | 49.4% acc., 40.1% f1 |
| | VGGNet | 48.2% acc., 38.7% f1 |
| | AlexNet | 41.8% acc., 34.1% f1 |
| [67] | GoogleNet | 98.7% acc., 97.1% f1 |
| | AlexNet | 99.2% acc., 98.5% f1 |
| [68] | AlexNet | 99.4% acc. |
| | VGG16 | 99.5% acc. |
| [63] | VGG16 | 60.4% acc., 60.0% f1 |
| | InceptionResNet V2 | 70.5% acc., 70.0% f1 |
| | Inception V3 | 62.1% acc., 61.0% f1 |
| [82] | LeNet | 98.6% acc., 98.6% f1 |

As can be observed from Table 5, several CNN architectures developed over the last decade have been successfully used, showing great potential for agriculture applications. From these, the use of older CNN architectures such as AlexNet (2012), VGG16 (2014), and GoogleNet (2014) were found on 44%, 33%, and 33% of the analysed papers, respectively. Although the use of the most recent CNN architectures is not expressed in the papers analysed in this review, we believe that, in the near future, the application of newer architectures to agriculture will be a reality.

### 2.3.4. Transfer Learning

TF makes use of already existing knowledge for some related task or domain in order and apply it to the problem under study. Models previously trained for image classification on large data-sets are usually used and adapted to the data-set under study. A common approach is to substitute the last network layers (i.e., the dense layers) of a pre-trained network, adapting it for a different classification task. The model is then trained but only the newly inserted layers are trainable—all network layers remain frozen during the training process. In extension of this approach, fine-tuning, is also commonly used. Besides training the newly inserted layers, fine-tuning allows the training of additional layers of the base model, typically the deeper convolutional layers of the network.

TF is usually done when the studied data-set is small, with insufficient samples for training a CNN model from scratch.

Table 6 synthesizes several deep learning-based studies where TF was applied. It presents details addressing: the data-set used for the base model training, the used TF method and the performance difference between using TF and training from scratch.

**Table 6.** TF analysis.

| Study | Model | Dataset for Pretrain | Method | Performance Difference Compared to Training from Scratch |
|---|---|---|---|---|
| [13] | AlexNet, GoogleNet | ImageNet | All layers trainable | ~−2% acc. |
| [30] | CaffeNet | ImageNet | Low learning rate for original layers (0.1), high for top layer (10) | ~−0.50% acc. |
| [62] | AlexNet, GoogleNet, VGGNet, ResNet | ImageNet | Fine tune | ~−14.0% acc. in best model (ResNet) |
| [15] | Faster RCNN (ResNet101, Inception V2, Inception ResNet V2) | COCO | Fine tune | No comparison |
| [67] | AlexNet, GoogleNet | ImageNet | Fine tune | ~−2% |
| [63] | VGG16, Inception V3, Inception ResNet v2 | ImageNet and/or PlantVillage | Fine tune | ~−31.0% using ImageNet and PlantVillage |

As can be observed from the table, the use of TF leads to lower performance when compared with training the full model from scratch. Nevertheless, there are many cases where such a difference is small, which means that TF can indeed be a useful possibility when the data-set is not sufficiently large.

## 3. Discussion

The studies collected in this review show that plant disease classification is a domain with promising results, with some studies achieving very high results [13,67]. Diverse data-sets have been employed, each with their own characteristics and associated difficulties: intraclass variability, background diversity, and different lighting and shading conditions during image acquisition. Due to these reasons, performance comparisons between the analyzed studies is not a straightforward task.

### 3.1. Data Acquisition

The data acquisition phase will have great influence on the quality of the ML model results, since the quantity and quality of data will influence the behaviour of ML models both in terms of good results and robustness.

Disease classification has shown promising results when the images of leaves are taken in laboratory conditions, with a single leaf against a neutral color background [13]. Laboratory image data is usually acquired using controlled lighting conditions using neutral color image backgrounds. On the other hand, images acquired in the cultivation fields are much harder to classify due to the presence of multiple leaves and plants, varying shading and lighting conditions, different ground textures and background objects [63]. Training ML models using laboratory images does not transfer well to testing in field conditions, achieving poor results. As for the opposite, reasonable results have been achieved [63,68]. The inclusion of images acquired in the field to the training processes can greatly improve performance of the models [68]. Nevertheless, the choice of data should always reflect the target objective of the application. For instance, if the model is intended to work in the laboratory environment, it isn't necessary to train it using data acquired on uncontrolled environments, since robustness to different acquisition conditions will not be necessary for the case. On the other hand, if the model is to be used in the field, images from the real context must be used. If the latter is not ensured using a large amount of images, the model will probably not perform well on the real conditions that exist in the crop fields.

Many studies applied ML-based algorithms to weather data records, since the weather conditions have an important role in the development of diseases and pests. Sensors for measuring weather data such as temperature, humidity, and rainfall fill data records that can be used to find correlation between the data and the development of pests or plant diseases in order to predict theses kind of developments [14,28,29]. Data records of meteorological measurements and pest occurrence can also be analyzed by deep learning with good results, as can be seen in [14]. Pest occurrence can be periodically manually analyzed by counting insect appearances [32] or by automated processes using computer vision techniques that detect insects on leaves or in traps [15,28,71].

Plants reflect NIR radiation differently depending on the disease damage [52]. If remote sensing data [32] or cameras equipped with filters [40] (allowing the capture of images at different bands of the light spectrum) are available, they can be used to compute NDVI. In [32], Skawsang et al. finds the relevance of NDVI and temperature for predicting pest occurrence. In [32], it is suggested that NDVI contains information about the relation between the crop growth stage and pest development. Duarte-Carvajalino et al. suggests in [40] that the NIR band is more suited for late blight detection than color imagery.

Thermal imagery combined with colored images provides higher accuracy when compared with using color features only, when detecting diseases that affect the temperature of the plant [27]. This is especially useful for detecting specific diseases in their early stages, where the plant has not yet developed visible symptoms. Knowledge about the variables that will influence the state of the plantation being worked on is very important when it comes to deciding which types of data should be acquired.

### 3.2. Data Pre-Processing

Data pre-processing techniques vary according to the used ML-based approach. In the case of image data, feature extraction can be done manually by applying computer vision algorithms, or automatically using deep learning.

Manual feature extraction processes typically demands pre-processing steps such as noise reduction or contrast enhancement. The researchers have to decide and select which feature extractors are more suitable for the problem at hand. When using deep learning, pre-processing is typically focused on data augmentation, enriching the training data-set in order to achieve a better model generalization. Deep learning shows better results when directly analyzing the originally acquired images when compared with the use of images converted to grey-scale [82] or subject to background removal [13]. This is a useful finding because background removal can be a complex and arduous task for images taken in field conditions, with complex and varying background [68]. When comparing the performance of ML models based on manual feature selection with models based on deep learning, the latter has shown better performance in studies that compared both approaches using the same input data [15,67,83].

Highlighting the region of interest of the leaves and reducing the background noise can increase the model's performance. This is valid for plant disease identification [63] as well as for insect classification [71].

### 3.3. Machine Learning Models

The studies presented along this paper have mostly used SVM, RF, or deep learning-based ML models. All of these have show promising results, highlighting the potential of using ML techniques for disease and pest classification, detection, and prediction. SVMs are robust and useful in high dimensional spaces due to their use of kernel trick. RF can avoid overfitting due to the high number of trees trained in different subsets of data. Deep learning usually achieves the best classification results due to its ability to create and extract hierarchical features from the inputs. Deep learning beats other ML models, particularly in image classification domains, especially when using pre-existing CNN architectures such as Inception and ResNet [62,63].

Despite deep learning models achieving higher accuracy values, SVM and RT can also achieve high values with accuracy above 94%, especially in disease classification on laboratory images [67]. SVM also achieves high accuracy, with values above 90%, in the detection of tomato diseases [30].

RNNs are capable of establishing relationships between weather data and pest occurrence, surpassing other models such as RF and SVM [14].

In scenarios where data is difficult to obtain, models trained with a lower amount of data can benefit from the use of TF, rather than having the models trained from scratch [13,52,72,76,77]. Most studies have their models pre-trained on large data-sets for image classification such as ImageNet or COCO. The inclusion of the PlantVillage dataset with ImageNet for pre-training helps to improve the accuracy of models for disease classification on images acquired in the field [62]. TF is typically applied by training some of the top layers of the pre-trained model jointly with the new classifier.

An alternative would be to address lack-of-data problems using few-shot learning approaches, as suggested in [89].

## 4. Conclusions

This survey presented an insight into existing research addressing the application of ML-based techniques for forecasting, detection, and classification of diseases and pests.

Data-sets containing weather, diseases, and pests data should keep records for long periods of time. Time-series ML models, such as RNN, can be employed to accurately forecast the occurrence of diseases and pests based on meteorological measurements series. NDVI measurements can also be helpful, since they provide additional information regarding the crop's development.

Detection and classification of pests and diseases can be performed using computer vision and deep-learning algorithms based on CNN models, which show better performance when compared with older image classification approaches based on "manual" features extraction. However, deep learning models require large amounts of data, which can be difficult to obtain. To tackle this issue, the use of transfer learning or few-shot learning methods can prove useful. Nonetheless, although the performance of deep learning-based methods is high for images acquired under controlled conditions, additional research is required regarding the analysis of images taken in the field, under real life conditions.

Since the literature does not yet include substantial work on pest and disease forecasting using combinations of different data modalities, this article also aimed to provide a general overview on the use of ML techniques over different types of data, in order to facilitate further developments that may help fulfill this gap.

**Author Contributions:** T.D.: research and writting; T.B.: writing and supervision; J.C.F.: supervision. All authors have read and agreed to the published version of the manuscript.

**Funding:** This research has received funding from the ECSEL Joint Undertaking (JU) under grant agreement No 876925.

**Institutional Review Board Statement:** Not applicable.

**Informed Consent Statement:** Written informed consent has been obtained from the patient(s) to publish this paper.

**Data Availability Statement:** Not applicable.

**Acknowledgments:** We would like to thank Inov-Inesc Innovation for the conditions it provided for the realization of this article. This project has received funding from the ECSEL Joint Undertaking (JU) under grant agreement No 876925. The JU receives support from the European Union's Horizon 2020 research and innovation programme and France, Belgium, Germany, Netherlands, Portugal, Spain and Switzerlan.

**Conflicts of Interest:** The authors declare no conflict of interest.

## Abbreviations

The following abbreviations are used in this manuscript:

| | |
|---|---|
| ANN | Artificial Neural Network |
| CNN | Convolutional Neural Network |
| GLCM | Grey Level Co-occurrence Matrix |
| HoG | Histogram of oriented Gradient |
| ILSVRC | Large Scale Visual Recognition Challenge |
| JU | ECSEL Joint Undertaking |
| KNN | K-Nearest Neighbor |
| LPB | Local Binary Pattern |
| LSTM | Long Short Term Memory |
| ML | Machine Learning |
| MLP | Multi-Layer Perceptron |
| NDVI | Normalized Difference Vegetation Index |
| NIR | Near Infra-Red |
| PETA | Progressive Environmental and Agricultural Technologies |
| RF | Random Forest |
| RNN | Recurrent Neural Network |
| SGD | Stochastic Gradient Descent |
| SIFT | Scale Invariant Feature Transform |
| SURF | Speeded Up Robust Features |
| SVM | Support Vector Machine |
| TF | Transfer Learning |
| TPMD | Tomato Powdery Mildew Disease |

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
