# Peer review of "Machine Learning for Detection and Prediction of Crop Diseases and Pests: A Comprehensive Survey"

_agriculture, doi:10.3390/agriculture12091350_

Round 1

Reviewer 1 Report

Machine Learning for Detection and Prediction of Crop Diseases and Pests: A Comprehensive Survey

The work should be classified as a review and not as a standard article.

The issue of artificial intelligence based on few shots learning has not been addressed ( see e.g.: doi: 10.1186/s13007-022-00866-2)

Also spectral imaging has been also marginally mentioned

It would be interesting to add also a section/paragraph mentioning what kind of tools have reached the market, and are available for farmers (or in other words which studies can be practically implemented)

I would change Figure 2: it looks to me too much simplicistic

The tables provide information for each paper: I would prefer having more quantitative representation (e.g. 80% of papers resize; 30% of papers do data augmentation;...) : in this way the reader might get an understanding of what is more studied. 

Throughout the paper only a tomato diseased plant leaf is shown. In a paper with such a topic I would report some more images of diseased plant leaves.

 In case the paper will be accepted for revision, please address above comments and correct accordingly the paper,

- giving your pertinent comments in the “Response to reviewer” document

- reporting in the “Response to reviewer” document also the paragraph with amended text highlighted with yellow colour or the new amended figure.

Author Response

Dear Reviewer,

Thank you for giving us the opportunity to submit a revised draft of our manuscript titled “Machine Learning for Detection and Prediction of Crop Diseases and Pests: A Comprehensive Survey” to Agriculture (ISSN 2077-0472).  We are grateful and appreciate your time and effort for providing us with valuable feedback on our manuscript. We incorporated changes that aim to reflect most of your comments and suggestions. We have highlighted those changes within the new version of the manuscript. Below you can find our responses to your comments:

Point 1: The work should be classified as a review and not as a standard article.

Response 1: Thank you for pointing this out. We configured the tex file in order to correctly display “review” instead of “article”.

Point 2: The issue of artificial intelligence based on few shots learning has not been addressed (see e.g.: doi: 10.1186/s13007-022-00866-2).

Response 2: We agree with the suggestion and addressed it in the manuscript. We analyzed the source kindly provided by the reviewer and added a brief concept explanation, the problem it mitigates, and a reference to the work for additional details. The additional text addressing this comment consists on two paragraphs, at the end of sections 2.3.3 and 3.3.

Point 3: Also spectral imaging has been also marginally mentioned.

Response 3: We addressed this comment by including additional references that demonstrate the utility of spectral imaging for computing vegetation indices, since these allow to obtain important information regarding the development status of the crop fields. Additional text and references were included in the third paragraph of section 2.1.

Point 4: It would be interesting to add also a section/paragraph mentioning what kind of tools have reached the market, and are available for farmers (or in other words which studies can be practically implemented).

Response 4: We included an additional paragraph near the end of the introduction, mentioning examples of tools and services currently available on the market.

Point 5: I would change Figure 2: it looks to me too much simplistic.

Response 5: Figure 2 was based on the one published in:

https://earthobservatory.nasa.gov/features/MeasuringVegetation/measuring_vegetation_2.php .

Since the figure was mostly used for illustrating the concept (different ratios between reflect/absorbed radiation depending on the plant’s health), we opted to keep it simple. Nevertheless, we believe that the reviewer comment may have been motivated by the lack of explanations regarding the computations bellow the tree images additional explanations and therefore we updated the figure by removing them. We also modified the caption and included a reference to the webpage above, where additional information can be retrieved.

Point 6: The tables provide information for each paper: I would prefer having more quantitative representation (e.g. 80% of papers resize; 30% of papers do data augmentation;...) : in this way the reader might get an understanding of what is more studied.

Response 6: We included text providing the most relevant quantitative information regarding the analyzed work depicted in tables 2 and 5. For Table 3 (SVM algorithms) we included an additional sentence stating that linear, polynomial and RBFs kernel use were the most commonly found in literature.

Point 7: Throughout the paper only a tomato diseased plant leaf is shown. In a paper with such a topic I would report some more images of diseased plant leaves.

Response 7: To tackle this issue, we included additional images regarding insect pests and plant diseases. Some of these were include as illustrations associated to image processing algorithms. This inclusion was performed in sections 2.1.2, 2.2.1 and 2.2.2 (a total of six new images).

Reviewer 2 Report

Reviewer’s Comments

The manuscript " Machine Learning for Detection and Prediction of Crop Diseases and Pests: A Comprehensive Survey," with the Manuscript id: agriculture-1860035, is very well written. Although I recommend accepting the manuscript, I do have a few comments and suggestions that should be made before it is officially accepted.

following are the comments:

1.      Author must check for unnecessary capitalization and try to avoid it.

2.      Line no 93-94: check the grammar of the sentence, “With this in mind, in the following sections various works and the respective techniques used in the different parts of the pipeline will be discussed.”, please rewrite it.

3.      Line 127: remove repeated, “the” in the first sentence.

4.      Line 351-353: Rewrite the sentence, “SURF is conceptually similar to SIFT, with advantage of being much faster, which is may be relevant for the implementation of real-time applications.”, avoid the use of may.

5.      Line 498: The author can add more information about the LSTM and its application in the field of review.

6.      Line 562: Check for “It The”, please correct it.

7.      As the manuscript is well written, but the conclusion seems to be the extension of the discussion, the author should rewrite the conclusion, and the major finding must be presented in max of 200 words.

I wish authors a great success.

Author Response

Dear Reviewer,

Thank you for giving us the opportunity to submit a revised draft of our manuscript titled “Machine Learning for Detection and Prediction of Crop Diseases and Pests: A Comprehensive Survey” to Agriculture (ISSN 2077-0472).  We are grateful and appreciate your time and effort for providing us with valuable feedback on our manuscript. We incorporated changes that aim to reflect most of your comments and suggestions. Apart from the smaller glitches, we highlighted the changes within the new version of the manuscript. Below you can find our responses to your comments:

Point 1: Author must check for unnecessary capitalization and try to avoid it.

Response 1: Thank you for this suggestion. We revised the article considering your comment and therefore we hope to have found and correct all cases of unnecessary capitalizations.

Point 2: Line no 93-94: check the grammar of the sentence, “With this in mind, in the following sections various works and the respective techniques used in the different parts of the pipeline will be discussed.”, please rewrite it.

Response 2: Thank you for pointing this out. We rebuilt this sentence in order to correct its grammar. It now stands as follows: “The following sections present and discuss different approaches used in these three stages.”.

Point 3: Line 127: remove repeated, “the” in the first sentence.

Response 3: Thank you for pointing out this glitch. It was corrected.

Point 4: Line 351-353: Rewrite the sentence, “SURF is conceptually similar to SIFT, with advantage of being much faster, which is may be relevant for the implementation of real-time applications.”, avoid the use of may.

Response 4: Thank you for the suggestion and for pointing out the glitch. We rewritten the sentence as “SURF is conceptually similar to SIFT, with the advantage of being much faster, which can be relevant for the implementation of real-time applications.”.

Point 5: Line 498: The author can add more information about the LSTM and its application in the field of review.

Response 5: Thank you for this suggestion. To address this suggestion, we included two additional paragraphs in section 2.3.3: one about LSTM-based networks, which tries to briefly explain the concept, and another regarding an application to the agriculture context.

Point 6: Line 562: Check for “It The”, please correct it.

Response 6: Thank you for pointing this out. We corrected the glitch and performed small adjustments to the sentence. It now reads “It presents details addressing: the data-set used for the base model training, the used TF method and the performance difference between using TF and training from scratch.”.

Point 7: As the manuscript is well written, but the conclusion seems to be the extension of the discussion, the author should rewrite the conclusion, and the major finding must be presented in max of 200 words.

Response 7: Thank you for this suggestion. We rewritten the conclusion to make it substantially shorter and concise. It remains slightly above the 200 word limit, but we hope it to be aligned with the reviewer’s suggestion.

Round 2

Reviewer 1 Report

Machine Learning for Detection and Prediction of Crop Diseases and Pests: A Comprehensive Survey

The paper has been clearly improved after the revision process. I have just two minor amendments to recommend: 

- in figures 5 and 6 I would insert the original image on the left and the processed image on the right

- in table 5 I would use always one decimal digit for percentages higher than 10%, and would usee to decimal digits for numbers lower than one. 

Author Response

Dear Reviewer,

Thank you for giving us the opportunity to resubmit a revised draft of our manuscript titled “Machine Learning for Detection and Prediction of Crop Diseases and Pests: A Comprehensive Survey” to Agriculture (ISSN 2077-0472).  We are grateful and appreciate your time and effort for providing us with valuable feedback on our manuscript. We incorporated changes that aim to reflect your comments and suggestions. We have highlighted those changes within the new version of the manuscript. Below you can find our responses to your comments:

Point 1: in figures 5 and 6 I would insert the original image on the left and the processed image on the right.

Response 1: Thank you for pointing this out. We configured the figures in order to display the images as suggested.

Point 2: in table 5 I would use always one decimal digit for percentages higher than 10%, and would usee to decimal digits for numbers lower than one.

Response 2: We agree with the suggestion and addressed it in the manuscript. We rewrite the values in this table and in the others according to the suggestion given.
